# Persistent accelerations disentangle Lagrangian turbulence

Lukas Bentkamp[1,2], Cristian C. Lalescu [1] & Michael Wilczek [1,2]

Particles in turbulence frequently encounter extreme accelerations between extended periods of quiescence. The occurrence of extreme events is closely related to the intermittent spatial distribution of intense flow structures such as vorticity filaments. This mixed history of flow conditions leads to very complex particle statistics with a pronounced scale dependence, which presents one of the major challenges on the way to a non-equilibrium statistical mechanics of turbulence. Here, we introduce the notion of persistent Lagrangian acceleration, quantified by the squared particle acceleration coarse-grained over a viscous time scale. Conditioning Lagrangian particle data from simulations on this coarse-grained acceleration, we find remarkably simple, close-to-Gaussian statistics for a range of Reynolds numbers. This opens the possibility to decompose the complex particle statistics into much simpler sub-ensembles. Based on this observation, we develop a comprehensive theoretical framework for Lagrangian single-particle statistics that captures the acceleration, velocity increments as well as single-particle dispersion.

[1] Max Planck Institute for Dynamics and Self-Organization, Am Faßberg 17, 37077 Göttingen, Germany. [2] Faculty of Physics, University of Göttingen, Friedrich-Hund-Platz 1, 37077 Göttingen, Germany. Correspondence and requests for materials should be addressed to M.W. (email: michael.wilczek@ds.mpg.de)

Complex systems, which feature many excited degrees of freedom and strong nonlinear interactions, notoriously defy a reductionist approach. Statistically, these features imply multi-scale correlations and significant departures from Gaussianity. Turbulence, the disordered state of a strongly driven fluid, is a paradigm for this class of systems. Stirred on large scales, kinetic energy is passed on to ever smaller scales until dissipated into heat. Cascading flow instabilities generate intense small-scale vortices and dissipation events with an intermittent spatial distribution. This spatial intermittency leads to extended phases of quiescence interrupted by episodes of violent accelerations along individual trajectories of tracer particles, as they probe turbulence in space and time. Statistically, this implies a pronounced scale dependence, with extreme fluctuations on small temporal scales[1–6].

A recurrent idea of how to advance the understanding of these systems is the statistical reduction of complexity. It underlies various modeling approaches for complex systems, and in particular turbulence, such as superstatistics[7–13] and multifractals[14–20]. In these approaches, the full ensemble statistics is obtained by superposing simpler statistics characterized by a statistically distributed parameter. In superstatistics, for example, the sub-ensembles are assumed to be in thermodynamic equilibrium at fixed temperatures, but the temperature fluctuates across the ensemble. To obtain a physically meaningful description for a complex system like turbulence, the challenge is to identify the quantity that separates the full statistical ensemble into simpler sub-ensembles. Such a quantity has so far remained elusive for Lagrangian turbulence.

For the description of the spatial (i.e. Eulerian) statistics of turbulence, Kolmogorov and Obukhov[21,22] introduced the idea that intermittency is rooted in strong spatial fluctuations of the dissipation rate. In their classic theory, the Eulerian refined similarity hypothesis states that the scale-dependent Eulerian statistics can be reduced to a universal form, solely depending on the dissipation rate volume-averaged (i.e. coarse-grained) on the given scale. Indeed, conditional probability density functions (PDFs) of velocity increments were observed to take a self-similar, or even an approximately Gaussian form when conditioning on the coarse-grained energy dissipation[23,24] or energy transfer rate[25,26].

For Lagrangian turbulence, similar ideas have been proposed, referred to as the Lagrangian refined similarity hypothesis[27–31]. Here, it is assumed that Lagrangian statistics can be formulated in a universal form in terms of a locally averaged (in time or space) dissipation rate along tracer particle trajectories. However, conditional statistics appear to remain intermittent and generally depart from Gaussianity[23,32]. This raises the question whether there exists a physically meaningful quantity that separates the intermittent statistics of Lagrangian turbulence into simpler, Gaussian statistics.

In this paper, we provide an answer to this question. We introduce the squared acceleration, coarse-grained over a typical viscous time scale, as a measure of persistence of the Lagrangian acceleration, and show that it decomposes the strongly non-Gaussian and scale-dependent statistics of Lagrangian turbulence into much simpler, close-to-Gaussian sub-ensembles. Based on this observation, we develop a comprehensive theoretical framework of Lagrangian single-particle statistics.

## Results

**Coarse-grained Lagrangian acceleration.** The central quantity of this work is the coarse-grained squared acceleration:

$$\alpha(t) \equiv \int_{-\infty}^{\infty} d\tau \, F_{\Theta}(\tau) \, \mathbf{a}^2(t + \tau). \tag{1}$$

Here, $\mathbf{a}(t)$ denotes the acceleration vector along a Lagrangian trajectory and $F_{\Theta}(\tau)$ a Gaussian filter kernel with standard

deviation $\Theta$. This time-averaged squared temporal derivative of the Lagrangian velocity can be perceived as a Lagrangian analog of the volume-averaged squared spatial derivatives of the velocity field (i.e. the coarse-grained dissipation rate) in the Eulerian refined similarity hypothesis[21,22], which play a key role in separating Eulerian statistics into simpler sub-ensembles. Contrary to the refined similarity hypothesis, where the coarse-graining scale varies with the scale under consideration, we here fix the coarse-graining time scale $\Theta$. The resulting quantity $\alpha$ measures the degree of small-scale turbulence encountered by a particle. While in principle also the instantaneous acceleration would be a suitable quantity to achieve this, a coarse-graining is needed to determine if the acceleration persists in time and can be considered representative of the local flow. We therefore call this coarse-grained squared acceleration the (squared) persistent acceleration.

If the coarse-graining time scale is too small, we rely on a criterion based on a single instant in time, which may not represent the flow region well. On the other hand, if the coarse-graining time scale is too large, Lagrangian tracers will probe regions of mild and strong turbulence within the same coarse-graining window, reducing the informative value of $\alpha$. Our results suggest that, for the Reynolds number range under consideration, a reasonable choice of $\Theta$ is given by 2 or $3\tau_\eta$ ($\tau_\eta$ is the Kolmogorov time scale). This corresponds roughly to the time scale on which Lagrangian acceleration components are significantly correlated; in fact, autocorrelation functions of acceleration components have been observed to cross zero at $\sim 2\tau_\eta$, almost independent of Reynolds number[1,33–35]. More information on the choice of $\Theta$ along with a study of the sensitivity of our results can be found in Supplementary Note 1.

As shown in Fig. 1, intense vorticity filaments exert significant centripetal acceleration on tracer particles, which can last for several Kolmogorov time scales[36–39]. A key observation is that

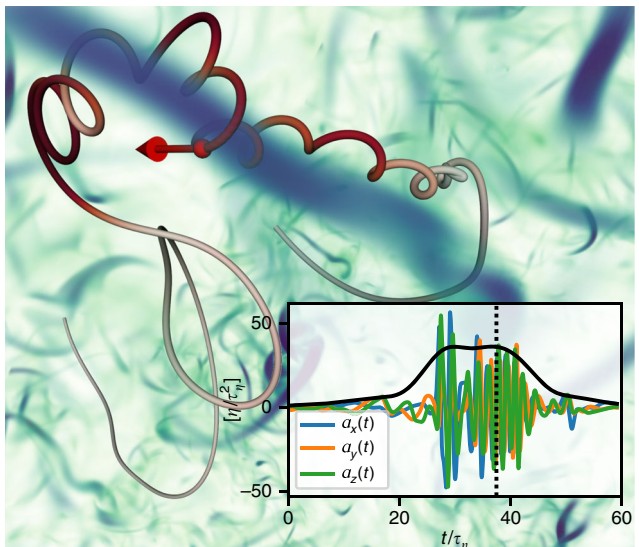

**Fig. 1** Tracer particle encountering a vortex filament in turbulence. The tracer trajectory is colored according to its instantaneous acceleration magnitude, and the blue-green volume-rendering corresponds to the intensity of the vorticity field. The particle acceleration components oscillate strongly in time (inset, in Kolmogorov units) when encountering the intense vortex filament. The root of the squared acceleration, coarse-grained over a few Kolmogorov time scales, varies only weakly during such an event (inset, black curve). The dashed line indicates the instant in time at which the vorticity field is visualized and the tracer is rendered as a sphere

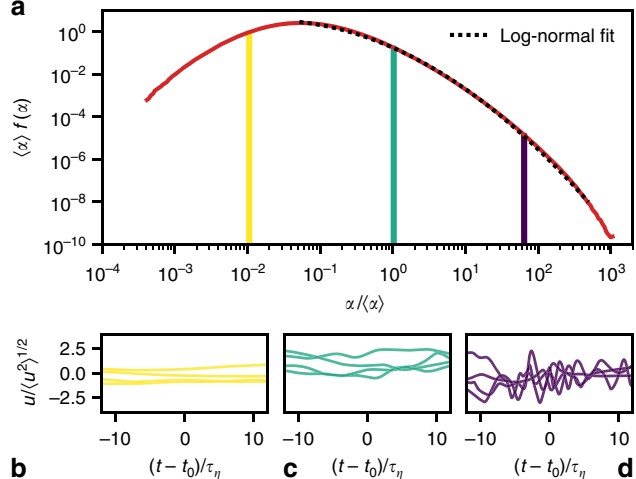

**Fig. 2** Distribution of $\alpha$ and velocity time series for various values of $\alpha$. **a** The PDF of $\alpha$ for $\Theta = 3\tau_\eta$; the right tail of the PDF is accurately fitted by a log-normal distribution (dotted line). **b**–**d** Examples of velocity components $u(t)$ for trajectories with fixed values of $\alpha(t_0)$ (the corresponding bins are highlighted in panel **a**, and colors correspond to the ones in Figs. 3 and 4). Oscillations of the Lagrangian velocity components increase with $\alpha$

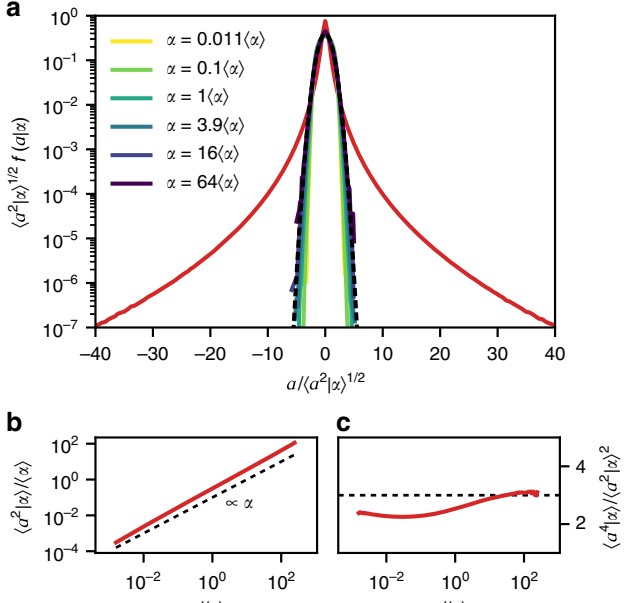

**Fig. 3** Lagrangian acceleration statistics decomposed into Gaussian sub-ensembles. **a** The PDFs $f(a|\alpha)$ of acceleration components conditional on $\alpha$ (colored lines) are all close to Gaussian. The unconditional PDF (red line) and a Gaussian distribution (black, dashed line) are plotted for comparison. **b** The conditional second-order moment of the acceleration increases slightly faster than linearly. **c** The flatness of the conditional acceleration is close to the Gaussian value three for a large range of $\alpha$. The conditional ensemble average on $\alpha$ is denoted by $\langle \cdot | \alpha \rangle$

particles retain high values of $\alpha$ during close encounters with these intense small-scale structures (sometimes termed particle-trapping events). For the trajectory visualized in Fig. 1, for example, the value of $\alpha$ is more than a hundred times its mean. High values of the coarse-grained acceleration are therefore an indication that Lagrangian tracer particles probe strongly turbulent flow regions. In contrast to that, low values of $\alpha$ are expected during episodes in which particles drift through comparably quiescent flow. A quantity similar to $\alpha$, the time-averaged acceleration magnitude, has already been found to be a good discriminator for Lagrangian intermittency in previous literature. By filtering out events above a certain threshold of this quantity, Biferale et al.[38] showed that vortex trapping significantly affects Lagrangian small-scale intermittency.

Pursuing the idea that Lagrangian intermittency can be disentangled based on the local intensity of the small-scale turbulence, in the following we discriminate Lagrangian trajectories from direct numerical simulation (DNS) data by means of the coarse-grained squared acceleration $\alpha$. To test our approach, we have investigated a comprehensive set of simulations of fully developed turbulence in the Reynolds number range (based on the Taylor microscale $\lambda$) $R_\lambda \in [210, 509]$ with up to 16 million tracer particles (see Methods for details). In the following, we mainly focus on a well-resolved data set at $R_\lambda \approx 350$. Ensemble averages, denoted by $\langle \cdot \rangle$, are taken over the set of particles and additionally in time.

We first show the PDF of $\alpha$ with $\Theta = 3\tau_\eta$ in Fig. 2a. We find that the coarse-grained squared acceleration $\alpha$ scatters more than six orders of magnitude in terms of its mean. Note that this mean is identical to the acceleration variance, $\langle \alpha \rangle = \langle \mathbf{a}^2 \rangle$, as can be readily seen from Eq. (1). The resulting broad distribution can be interpreted as a signature of the spatio-temporal intermittency of acceleration. In previous literature[12,33] it was observed that the tail of the PDF of the instantaneous acceleration magnitude is well fitted by a log-normal distribution. As demonstrated by the fit in Fig. 2a, we find that this also holds for the coarse-grained squared acceleration. We note in passing that a fit over the entire range of $\alpha$ values can be achieved by an interpolation between an algebraic increase for small $\alpha$ and the log-normal decay for larger

$\alpha$ (not shown). Figs. 2b–d give an impression of typical velocity components $u(t)$ along Lagrangian trajectories for different values of $\alpha$. At very low values, the velocity is quasi-constant, corresponding to unperturbed, inertial motion; particles are essentially swept with the large-scale flow. For an average $\alpha$, the velocity varies slowly over time with some fluctuations, as they occur in mildly turbulent flow regions. This changes dramatically for high values of $\alpha$: here, particles undergo fast velocity oscillations. Such oscillations occur when tracer particles encounter intense vortices.

**Conditional statistics.** Next, we demonstrate that conditioning on $\alpha$ leads to remarkably simple, close-to-Gaussian statistics. As an extreme example, we choose the PDF of the strongly non-Gaussian Lagrangian acceleration, for which events up to several hundred standard deviations have been observed[3,18,40,41]. The heavy tails of this distribution are indicative of the frequent occurrence of extreme events, which play an important role, for example, in the context of cloud microphysics[42]. Fig. 3a shows the PDFs of an acceleration component $a$ (all components have identical statistics due to isotropy) both unconditional (red line) and conditional on $\alpha$ (colored lines). Whereas the unconditional PDF is extremely heavy-tailed, the conditional acceleration PDFs display a close-to-Gaussian form for all values of $\alpha$. Note that they have approximately zero mean and are shown in standardized form. Their variance, displayed in Fig. 3b, grows almost linearly with $\alpha$. As a quantitative benchmark, their flatness is shown in Fig. 3c. It is close to the Gaussian value three for all values of $\alpha$, with a tendency to sub-Gaussianity for low values of $\alpha$. We find the same behavior for multi-scale quantities like the velocity increment, which is presented in detail in Supplementary Note 2. Thus, the discrimination of Lagrangian trajectories by means of $\alpha$ appears to separate the statistical ensemble of Lagrangian

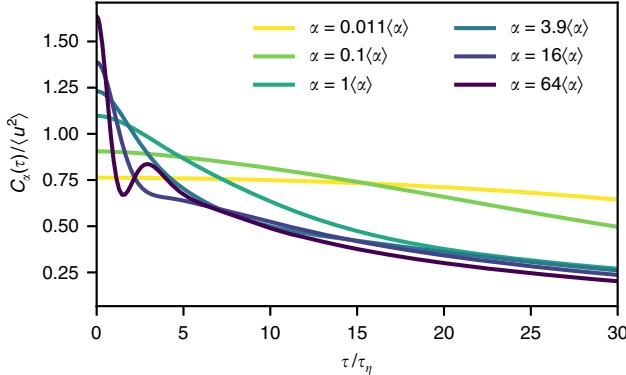

**Fig. 4** Conditional velocity autocorrelation functions. The conditional correlation functions on $\alpha$ reflect the strongly varying flow conditions for different values of $\alpha$. The higher $\alpha$, the faster the velocity of tracer particles decorrelates. Velocity oscillations observed for high values of $\alpha$ in Fig. 2d result in a secondary maximum in the corresponding correlation function

trajectories into much simpler, close-to-Gaussian sub-ensembles —a strong reduction of the complexity of Lagrangian statistics.

**A framework for single-particle statistics**. Based on this observation, we now develop a comprehensive framework of Lagrangian single-particle statistics. The complete statistical information of a Lagrangian trajectory is contained in its characteristic functional, which has been introduced in Hopf's functional approach to turbulence[43]. It allows us to derive single-particle statistics ranging from particle dispersion to multi-time velocity or acceleration statistics[44,45]. For simplicity, we here restrict ourselves to a single Lagrangian velocity component. Exploiting our observation that conditioning on $\alpha$ leads to approximately Gaussian statistics, we now assume Gaussianity in each sub-ensemble, i.e. for each value of $\alpha$. Mathematically, this is described by Gaussian characteristic functionals $\phi_\alpha^G[\vartheta]$, which take the form[44,46]

$$\phi_\alpha^G[\vartheta] = \exp\left[-\frac{1}{2}\int_{-\infty}^{\infty} dt \int_{-\infty}^{\infty} dt'\, \vartheta(t)\, C_\alpha(t-t')\, \vartheta(t')\right]. \quad (2)$$

Here, $\vartheta(t)$ denotes a test function. We assume the sub-ensembles to be statistically stationary and to have zero mean (like the entire flow). Gaussianity then implies that the statistics of each sub-ensemble is entirely determined by its autocorrelation function. These conditional velocity autocorrelation functions $C_\alpha(\tau) = \langle u(t_0 - \tau/2) u(t_0 + \tau/2)|\alpha(t_0)\rangle$ can be determined from the DNS data and are shown in Fig. 4 for different values of $\alpha$. They provide insight into the typical temporal evolution of the velocity. For small values of $\alpha$, corresponding to more quiescent regions of the flow, the autocorrelations start from a small variance and then decay slowly. This statistical observation is consistent with the sample trajectories shown in Fig. 2b. For higher values of $\alpha$, we observe an initially high value with a fast short-time decay. While tracers experience higher-amplitude fluctuations, velocities also decorrelate more quickly, consistent with the larger variety of particle trajectory geometries shown in Fig. 2c. At the highest values of $\alpha$, we find another indication for particle trapping events: Here the conditional velocity autocorrelation function exhibits a second local maximum after a few Kolmogorov time scales, a typical feature of oscillatory motion at a well-defined frequency. This feature can also be inferred from the sample trajectories shown in Fig. 2d.

To obtain the characteristic functional for the full ensemble, we need to evaluate the superposition of the Gaussian characteristic

functionals weighted by the PDF of $\alpha$[44,45]:

$$\phi[\vartheta] = \int_0^\infty d\alpha\, f(\alpha)\, \phi_\alpha^G[\vartheta]. \quad (3)$$

This means that the full Lagrangian ensemble can be considered as a probabilistic mix of Gaussian sub-ensembles with varying correlations. It has been demonstrated in previous literature that a superposition of Gaussian PDFs can be successfully employed to model specific statistical quantities, such as Eulerian[20,47] or Lagrangian[20] velocity increment PDFs, as well as acceleration PDFs[12]. Our framework generalizes such approaches: Since the characteristic functional offers a comprehensive statistical description, the complete single-particle statistics of Lagrangian turbulence can be determined from our framework once the PDF $f(\alpha)$ and the conditional correlation functions $C_\alpha(\tau)$ are given. For the present results, we take these quantities directly from our DNS data without resorting to further modeling assumptions.

**Comparison of theoretical and simulation results**. Next, we compare simulation results of various aspects of Lagrangian single-particle statistics with results obtained from our theoretical framework. As a starting point, we focus on second-order statistics. For example, the velocity autocorrelation function can be obtained by taking two functional derivatives of Eq. (3),

$$C^u(\tau) = -\left[\frac{\delta^2 \phi[\vartheta]}{\delta\vartheta(t)\delta\vartheta(t')}\right]_{\vartheta=0} = \int_0^\infty d\alpha\, f(\alpha)\, C_\alpha(\tau), \quad (4)$$

where $\tau = t - t'$. The resulting correlation function $C^u(\tau)$ is simply the averaged conditional correlation function. Therefore, this quantity is correctly captured by design. More generally, all quantities that are kinematically related to the velocity autocorrelation function, such as the mean squared displacement $\langle R(\tau)^2\rangle$ and the acceleration autocorrelation function $C^a(\tau)$, are accurately captured as well. Fig. 5a shows the mean squared displacement, which characterizes Lagrangian single-particle dispersion. A transition from a ballistic regime to a diffusive regime is observed. The correlation functions of the velocity and acceleration are shown in Fig. 5b. Compared to slow decay of the velocity autocorrelation, the acceleration autocorrelation shows the characteristic zero-crossing at about $2\tau_\eta$. As expected, these quantities are captured by our framework.

The main challenge in capturing Lagrangian single-particle statistics is intermittency, which can be studied in terms of the statistics of the velocity increment $v = u(t + \tau/2) - u(t - \tau/2)$ taken over a time lag $\tau$. Intuitively, velocity increments characterize velocity fluctuations across a given time scale and therefore are well suited to probe the multi-scale nature of Lagrangian turbulence. Statistically, intermittency manifests itself in a pronounced scale dependence of the PDF $f(v; \tau)$ of Lagrangian velocity increments $v$ taken over a time lag $\tau$. It exhibits heavy tails for small time lags but relaxes to an almost Gaussian distribution for large time lags. This is why it constitutes a prime example for the lack of statistical self-similarity in turbulence. For short times, the velocity increment is proportional to the acceleration. With appropriate standardization, the short-time limit of the velocity increment PDF is therefore given by the single-point acceleration PDF $f(a)$, whereas its long-time limit is related to the single-point velocity PDF $f(u)$ (through a convolution). As detailed in the Methods section, these PDFs can be derived from our framework (3) and expressed as a function of the PDF $f(\alpha)$ and the conditional velocity autocorrelation functions $C_\alpha(\tau)$. For example, we can determine the

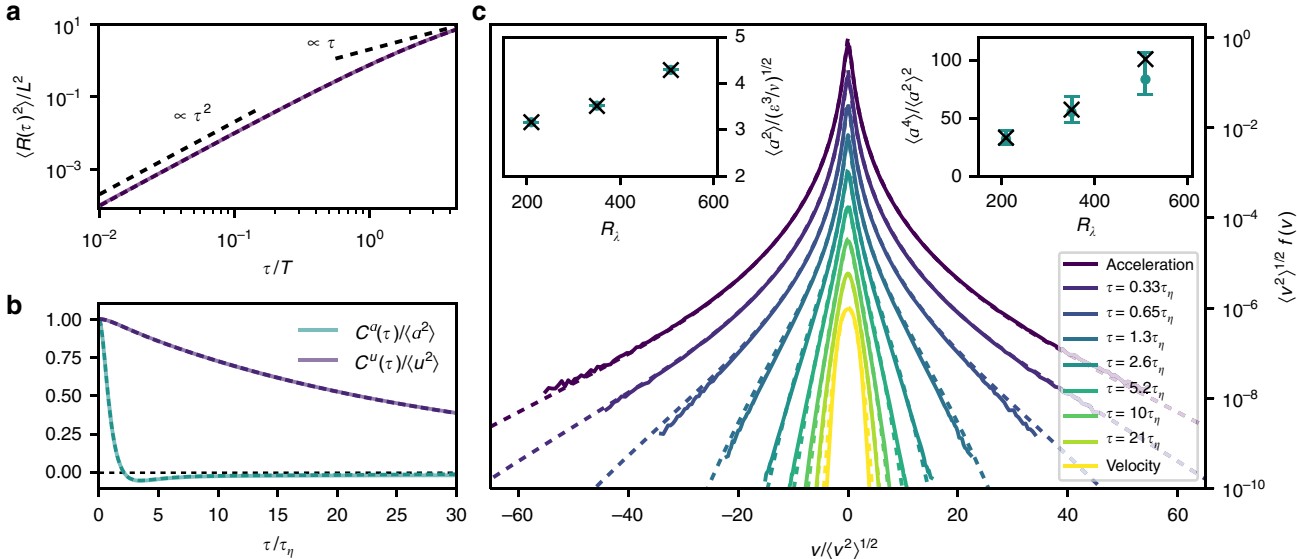

**Fig. 5** Comparisons of results from the characteristic functional framework with DNS data for $R_\lambda \approx 350$. **a** Mean squared displacement from the model (dashed) compared to DNS data (solid), given in Eulerian integral scales $L$ and $T$. The expected asymptotic ballistic and diffusive scalings are indicated for reference. **b** Velocity and acceleration correlation functions from the model (dashed) compared to DNS data (solid). **c** Comparison of PDFs derived from the model (dashed) with DNS data (solid). From top to bottom: acceleration PDF, velocity increment PDFs, velocity PDF (vertically shifted for clarity). For this comparison, we chose $\Theta = 3\tau_\eta$. Left inset: acceleration variance in Kolmogorov units from model (dots) compared to DNS (crosses). Right inset: acceleration flatness from model (dots) compared to DNS (crosses). The error bars indicate variations of the coarse-graining time scale $\Theta$ between $1.5\tau_\eta$ and $4.5\tau_\eta$.

increment PDF as

$$f(v; \tau) = \int\limits_0^\infty d\alpha\, f(\alpha)\, g(v; S_{2,\alpha}(\tau)). \quad (5)$$

Here, $g$ is a Gaussian increment distribution with the sub-ensemble second-order structure function $S_{2,\alpha}(\tau) = 2C_\alpha(0) - 2C_\alpha(\tau)$ as variance. Similarly, the velocity PDF is given by a superposition of Gaussians with variance $C_\alpha(0)$. The acceleration PDF is given by a superposition of Gaussian PDFs with variance $-C_\alpha''(0)$, where the double prime denotes a second derivative with respect to $\tau$.

As a quantitative benchmark of our framework, we compare the PDFs obtained from (3) to the ones obtained from DNS. Fig. 5c shows the velocity increment PDFs along with the PDFs of velocity and acceleration at $R_\lambda \approx 350$. Very good agreement is found at all scales, which supports our hypothesis that Lagrangian intermittency can be perceived as a consequence of the statistical mixture of different regions of the flow, each characterized by a particular coarse-grained squared acceleration. To test the Reynolds number dependence, we compute acceleration variance and flatness of the model and of the DNS for three different simulations in the range $R_\lambda \in [210, 509]$, which are shown in the insets of Fig. 5c. For all of them, we use the same coarse-graining time scale $\Theta = 3\tau_\eta$. By design, there is perfect agreement with the acceleration variance directly obtained from DNS data across all Reynolds numbers. Moderate deviations can be observed in the acceleration flatness. To test the sensitivity of our results with respect to the coarse-graining time scale, we vary it in the range $[1.5\tau_\eta, 4.5\tau_\eta]$. The resulting variation in acceleration flatness is indicated with error bars in the inset in Fig. 5c. For the simulation at $R_\lambda \approx 509$, we find optimal results for a smaller value of $\Theta$ at about $2\tau_\eta$. This points to a Reynolds-number dependence of the only free parameter in our framework. We address this issue in more detail in Supplementary Note 3. One important question for the next generation of experiments and simulations is whether an

asymptotic value of $\Theta$ can be reached at very high Reynolds numbers.

## Discussion

In conclusion, we showed that Lagrangian single-particle statistics can be decomposed into approximately Gaussian sub-ensembles when trajectories are discriminated with respect to the coarse-grained squared acceleration $\alpha$. Physically, high values of $\alpha$ correspond to events in which particles encounter intense small-scale structures such as vorticity filaments. Hence the decomposition intuitively separates regions of highly turbulent activity from more quiescent regions.

Based on this, we developed a theoretical model of Lagrangian single-particle statistics, which requires the PDF of $\alpha$ and the conditional Lagrangian velocity autocorrelation functions as input. Formulated in terms of the characteristic functional, our framework offers a comprehensive statistical description of Lagrangian single-particle statistics, which constitutes a conceptual generalization of previous approaches. By projecting it to finite-dimensional statistics, such as velocity increment distributions, we find very good agreement with simulation results. In particular, we find that our approach accurately captures Lagrangian intermittency across a range of Reynolds numbers.

Let us briefly comment on the implications of our findings for the systematic development of a predictive theory for Lagrangian turbulence. In this context, it is worth emphasizing that our framework provides more than just a model to fit one particular statistical quantity. It rather provides a comprehensive, self-consistent description of Lagrangian single-particle statistics: By design, it is consistent with any kinematic finite-dimensional statistical equation of Lagrangian turbulence. As an example, we remark that both the velocity increment PDF $f(v; \tau)$ and the mean acceleration conditioned on the increment $\langle a|v; \tau \rangle$ can be directly computed from the characteristic functional (3). It then can be shown (see Supplementary Note 4) that the kinematic evolution

**Table 1 Main DNS parameters**

| N | $R_\lambda$ | $\langle u^2 \rangle^{1/2}$ | L | $L/\eta$ | $T/\tau_\eta$ | $\frac{t_1 - t_0}{T}$ | n | $k_{max}\eta$ |
|---|---|---|---|---|---|---|---|---|
| 1024 | 210 | 1.05 | 1.06 | 144 | 19.5 | 23.7 | $2 \times 10^6$ | 3 |
| 2048 | 350 | 1.07 | 1.06 | 288 | 30.3 | 7.2 | $16 \times 10^6$ | 3 |
| 2048 | 509 | 1.05 | 0.99 | 549 | 47.9 | 7.4 | $16 \times 10^6$ | 1.5 |

Our simulations are run on three-dimensional periodic domains of side length $2\pi$ discretized on a real space grid with $N^3$ points over the time interval $[t_0,t_1]$. Along with the flow fields, $n$ tracer trajectories are integrated. Using the root-mean-squared velocity component $\langle u^2 \rangle^{1/2}$ and the energy spectrum $E(k)$, we define the integral length $L = \pi(\int dk\, E(k)/k)/(2\langle u^2 \rangle)$. The integral time scale is computed as $T = L\langle u^2 \rangle^{-1/2}$. The Kolmogorov length and time scales, $\eta$ and $\tau_\eta$ are computed from the mean kinetic energy dissipation $\varepsilon$ and the kinematic viscosity $\nu$. Based on the largest wavenumber $k_{max}$ resolved by our simulations, we compute the resolution criterion $k_{max}\eta$.

equation for the increment PDF[48]

$$\partial_\tau f(v; \tau) = -\partial_v [\langle a | v; \tau \rangle f(v; \tau)] \tag{6}$$

is satisfied. In this sense, our approach provides a self-consistent framework for Lagrangian single-particle statistics once the PDF of $\alpha$ and the conditional correlation functions are provided. For the current results, we obtained the PDF of $\alpha$ along with the conditional autocorrelation functions directly from DNS data. Once theoretical models for these quantities become available, our framework becomes fully predictive, which is an exciting direction for future work.

So far, theories of turbulence can be broadly categorized into predictive, but phenomenological models and rigorous, but unclosed (and therefore not predictive) approaches. By combining aspects of these two lines of research, our work helps to bridge this gap, which may lead to further theoretical progress in this long-standing problem.

## Methods

**Characteristic functional and reduced statistics**. The characteristic functional of the Lagrangian velocity time series $u(t)$ is defined as[46]

$$\phi[\vartheta] = \left\langle \exp\left( i \int_{-\infty}^{\infty} dt\, \vartheta(t) u(t) \right) \right\rangle, \tag{7}$$

where $\langle \cdot \rangle$ denotes an ensemble average and $\vartheta(t)$ a test function. Since it contains the full statistical information about the time series, we can derive arbitrary single-particle statistical quantities. Note that for presentational purposes, we choose the times $t_0$ and $t_0 + \tau$ for two-time quantities as opposed to the centered choice in the main text. Since the sub-ensembles are statistically stationary, this yields the same results.

Correlation functions can be computed from the characteristic functional in a straightforward manner. The Lagrangian velocity autocorrelation function $C^u(\tau)$ is obtained by taking functional derivatives:

$$C^u(\tau) = \langle u(t_0) u(t_0 + \tau) \rangle \tag{8}$$

$$\overset{(7)}{=} -\left[ \frac{\delta^2}{\delta\vartheta(t_0)\delta\vartheta(t_0+\tau)} \phi[\vartheta] \right]_{\vartheta=0} \tag{9}$$

$$\overset{(3)}{=} \int_0^\infty d\alpha\, f(\alpha)\, C_\alpha(\tau). \tag{10}$$

Various quantities are kinematically determined by this function. For instance, the acceleration autocorrelation function $C^a(\tau) = \langle a(t_0) a(t_0 + \tau) \rangle$ is given by its negative second derivative[49]:

$$C^a(\tau) = -\frac{d^2}{d\tau^2} C^u(\tau). \tag{11}$$

Let $X(t)$ denote one component of a Lagrangian trajectory. The mean squared displacement $\langle R(\tau)^2 \rangle = \langle (X(t_0+\tau) - X(t_0))^2 \rangle$ can be obtained by integration of the velocity autocorrelation function:

$$\langle R(\tau)^2 \rangle = \int_0^\tau dt \int_0^\tau dt'\, C^u(t - t'). \tag{12}$$

Finite-dimensional PDFs can be derived from Eq. (3) by an appropriate choice of the test function $\vartheta(t)$. In the simplest case, the single-point velocity statistics, we

set $\vartheta(t) = \gamma\delta(t - t_0)$, which yields the characteristic function

$$\phi^u(\gamma) = \langle \exp(i\gamma u(t_0)) \rangle \tag{13}$$

$$\overset{(7)}{=} \phi[\vartheta = \gamma\delta(t - t_0)] \tag{14}$$

$$\overset{(3)}{=} \int_0^\infty d\alpha\, f(\alpha) \exp\left( -\frac{\gamma^2}{2} C_\alpha(0) \right). \tag{15}$$

By a Fourier transform, we obtain the single-point velocity PDF

$$f(u) = \int_0^\infty d\alpha\, f(\alpha)\, g(u; C_\alpha(0)), \tag{16}$$

where $g$ is a Gaussian velocity distribution with variance $C_\alpha(0)$. Similarly, in order to obtain the characteristic function $\phi^a(\gamma)$ of the single-point acceleration, we choose $\vartheta(t) = -\gamma\frac{d}{dt}\delta(t - t_0)$, which yields

$$\phi^a(\gamma) = \langle \exp(i\gamma a(t_0)) \rangle \tag{17}$$

$$\overset{(7)}{=} \phi[\vartheta = -\gamma\frac{d}{dt}\delta(t - t_0)] \tag{18}$$

$$\overset{(3)}{=} \int_0^\infty d\alpha\, f(\alpha) \exp\left( \frac{\gamma^2}{2} C_\alpha''(0) \right). \tag{19}$$

From Eq. (17) to (18) and from Eq. (18) to (19) we have used integration by parts to swap the time derivative from the velocity to the delta function and from the delta function to the correlation function, respectively. The double prime denotes the second derivative with respect to $\tau$. Note that $C_\alpha''(0)$ is negative. The single-point acceleration PDF reads

$$f(a) = \int_0^\infty d\alpha\, f(\alpha)\, g(a; -C_\alpha''(0)). \tag{20}$$

Finally, the characteristic function $\phi^v(\gamma; \tau)$ of velocity increments over a time lag $\tau$ can be calculated by inserting $\vartheta(t) = \gamma\delta(t - t_0 - \tau) - \gamma\delta(t - t_0)$:

$$\phi^v(\gamma; \tau) = \langle \exp(i\gamma(u(t_0 + \tau) - u(t_0))) \rangle \tag{21}$$

$$\overset{(7)}{=} \phi[\vartheta = \gamma\delta(t - t_0 - \tau) - \gamma\delta(t - t_0)] \tag{22}$$

$$\overset{(3)}{=} \int_0^\infty d\alpha\, f(\alpha) \exp\left( -\frac{\gamma^2}{2} S_{2,\alpha}(\tau) \right). \tag{23}$$

Here $S_{2,\alpha}(\tau) = 2C_\alpha(0) - 2C_\alpha(\tau)$ is the sub-ensemble second-order structure function. Hence, the increment PDF is given by

$$f(v; \tau) = \int_0^\infty d\alpha\, f(\alpha)\, g(v; S_{2,\alpha}(\tau)). \tag{24}$$

In Fig. 5c, we compare second-order and fourth-order acceleration moments obtained from our framework to DNS data. They can be explicitly derived, for example, from the single-point acceleration PDF:

$$\langle a^2(t_0) \rangle = \int_{-\infty}^{\infty} da\, f(a)\, a^2 \tag{25}$$

$$\overset{(20)}{=} \int_0^\infty d\alpha\, f(\alpha)(-C_\alpha''(0)) \tag{26}$$

and

$$\langle a^4(t_0) \rangle = \int_{-\infty}^{\infty} da\, f(a)\, a^4 \tag{27}$$

$$\overset{(20)}{=} \int_0^\infty d\alpha\, 3f(\alpha)\, (C_\alpha''(0))^2. \tag{28}$$

**Description of DNSs**. To obtain high Reynolds number simulation data, we use a pseudo-spectral solver for the Navier–Stokes equations in the vorticity formulation with a third-order Runge–Kutta method for time stepping and a high-order Fourier smoothing[50] to reduce aliasing errors. The flow is forced on the large scales by maintaining a fixed energy injection rate in a discrete band of small Fourier modes $k \in [1.0, 2.0]$ (DNS units). Along with the flow field, we integrate tracer

trajectories using a second-order Adams–Bashforth method coupled to a first-order spline interpolation which is computed over a kernel of $8^3$ grid points (as detailed in ref. [51]). The characteristics of the DNS are summarized in Table 1. For the visualization in Fig. 1, data from a distinct simulation at $R_\lambda \approx 150$ was used.

## Data availability

The data that support the findings of this study are available from the corresponding author on request.

## Code availability

The simulation and post-processing codes that have been used to produce the results of this study are available from the corresponding author on request.

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

## Acknowledgements

We would like to acknowledge interesting and useful discussions with Laura J. Lukassen, and thank Dhawal Buaria for carefully reading the manuscript. The authors gratefully acknowledge the Gauss Centre for Supercomputing e.V. (www.gauss-centre.eu) for funding this project by providing computing time on the GCS Supercomputer Super-MUC at Leibniz Supercomputing Centre (www.lrz.de). Computational resources were also provided by the Max Planck Computing and Data Facility. The authors would like to thank Bérenger Bramas and Markus Rampp from the Max Planck Computing and Data Facility for the optimized particle tracking module used in our DNS, as well as general technical support. The visualization in Fig. 1 was generated with VTK[52]. This work was supported by the Max Planck Society.

## Author contributions

All the three authors made significant contributions to this work. L.B. contributed to the theoretical framework and analyzed the data. C.C.L. conducted the simulations and analyzed the data. M.W. designed the study and developed the theoretical framework. All authors wrote the paper together.

## Additional information

**Competing interests:** The authors declare no competing interests.

