## [Peer Review File · Nature Communications]

Reviewers' comments:

Reviewer #1 (Remarks to the Author):

The paper investigates the problem of intermittency observed in turbulent flows along Lagrangian particle trajectories. Understanding the Lagrangian nature of turbulence is a fundamental problem in the field, with a wide range of applications such as modeling chemical combustion, describing nutrient transport in the ocean, predicting pollutant dispersal in the atmosphere, etc

The paper can be viewed as providing a Lagrangian analogue of Kolmogorov's Refined Similarity Hypothesis framework, which is primarily an Eulerian description. Much of our current understanding of turbulent flows rests on Eulerian frameworks. There is widespread belief in the turbulence community that the underlying physics is ultimately Lagrangian in nature, yet a self-consistent and methodical translation of the main Eulerian results in turbulence to a Lagrangian formulation has remained elusive. The current work is an important contribution in this regard, which I believe is a promising framework to pursuing such a research program.

While I believe the paper is suitable for publication in Nature Communications, I have a few comments and questions, some of which I believe are important.

I feel that the presentation in its current form is perhaps too specialized and does not cater to the wide background of the journal's readers. I would suggest the authors to revise their manuscript by focusing more on the physical implications of their work, dwell less on the mathematical technicalities/details, and spend more effort at explaining the relevance of various plots. I would suggest they consider a reader who works in turbulence but is perhaps not versed in the techniques used by the authors: why should s/he care about PDFs of increments, their heavy tails, log-normal distributions, velocity correlations, characteristic functionals, and discuss more the physical insight afforded by the framework/model put forth by this work. For example, the first paragraph on page 4 illustrates how the paper can be more technical than is necessary. I believe these revisions will eventually benefit the paper and make it accessible to a wider audience. Some of my comments/questions below provide some more concrete suggestions.

Another important aspect of the paper which I would like the authors to address pertains to the coarse-graining time scale. What is special about $\sim 3-4$ Kolmogorov times? In Kolmogorov's refined similarity framework, the coarse-graining length scale is that of the scale being analyzed. Here, it does not seem related to any of the quantities being analyzed (e.g. in Fig. 4c). Why is that?

Other questions:

1) How persistent is the "persistent acceleration" put forth by the authors? Fig. 1 suggests it persists over ~ 10 Kolmogorov times. Is this the case?

2) The quantity α is introduced without much motivation. The reader has to wait until the last paragraph of the paper to see that it is analogous to the measure of dissipation. If this is what inspired the authors to focus on α , it would benefit the reader to discuss it soon after/before α is introduced in eq. (1).

3) The angular brackets first appear in Fig. 2 without being defined. The appendix defines them as an ensemble average. Please mention that in the main paper. In the appendix, mention how the ensemble average is taken in practice. Is it a time average and, if so, how do you justify/ensure statistical independence?

4) In Fig. 2a, it would help if the velocity time series corresponds to the same α values used in Fig. 2b and Fig. 3. Currently, Fig. 2a only shows velocity at 3 different values of α but each value shows 4 different u-series. Why is that and does the color legend correspond to that in Fig. 2b?

5) On page 3, bottom of the left column, the authors say "At high values of α , conditional velocity correlation functions exhibit a second local maximum after a few Kolmogorov time scales, which indicates oscillatory motion." The statement is somewhat vague, at least to me. I would expect oscillatory motion for any α , perhaps at varying amplitudes. This certainly appears to be the case in Fig. 1a. So how am I to interpret the secondary maximum in the conditional velocity correlation function for $\alpha=61$ but not at lower values of α ?

6) On page 4, the statement: "... Lagrangian intermittency can be perceived as a consequence of the statistical mixture of different regions of the flow, each characterized by a particular persistent acceleration." I believe this is insightful, but I have a few questions. First, is the use of the word "persistent acceleration" justified (related to question 1 above)? Over what time periods is this persistence observed and how does it compare to the coarse-graining time scale? Second, why is there sensitivity to the coarse-graining time scale and how should I interpret this dependence (related to the main comment above)? It is not clear if the coarse-graining time scale should be made longer or shorter as the Reynolds number is increased. Is it possible to present a plot of the optimal filter time scale as a function of Re ?

7) There is a typo in a sentence in the last paragraph of page 5. Too many "but".

Hussein Aluie

Reviewer #3 (Remarks to the Author):

This work proposes a framework to reduce the strongly non-Gaussian statistics of Lagrangian turbulence using sub-ensembles to Gaussian statistics and to predict from this statistical quantities. More specifically, the authors show that the acceleration probability functions become Gaussian once they are conditioned on an averaged acceleration. Next, using Gaussian characteristic functionals of this conditioned quantity they reconstruct the full characteristic functional that allow them to correctly predict for example the complete velocity increment statistics.

It is true that understanding the intermittent nature of turbulent flows is still an open problem. It is also true that Kolmogorov's and Obukhov's refined similarity hypothesis was an important milestone from the Eulerian point of view. Their coarse-grained energy dissipation rate disentangles sets with different scaling properties. The present manuscript proposes a mechanism to achieve the same from the Lagrangian point of view and to build on this a formalism to predict turbulence properties.

The present paper is well written and well structured. My concern about the present paper is that the proposed framework appears as a clever modelling although the authors explicitly reject this kind of criticism. The main quantity called 'persistent acceleration' α is the acceleration averaged along a tracer trajectory over a short time lag. This time lag is of the order of τ_η which is the correlation time of the tracer acceleration. Indeed, the authors find Gaussian statistics for the tracer acceleration once they condition it on its short time average. However, the condition α almost determines the acceleration (one point quantity) as the first is the (correlated) time average of the latter. Therefore all sets of

acceleration (for different α) will have a very small standard deviations. The subsequent step of the proposed framework, the prediction of turbulence properties such as velocity increment statistics uses measured quantities (from they direct numerical simulations) such as the occurrence of α and the velocity auto-correlation. So that the these predictions resemble reconstructions.

The authors should discuss in more detail the crucial conditioning on a short time lag. What happens for instance if the time lag tends to zero?

From a presentational point of view it would be instructive to present the flatness of the pdfs of the conditioned acceleration in FIG. 2 as a measure of the deviation from Gaussianity as a function of α .

Reviewer #4 (Remarks to the Author):

This is a pretty and nicely written article, that attacks a difficult emblematic problem of non-equilibrium non-linear physics. It is a real pleasure to read the article, and references give a pretty much complete overview of the field. Figures are clear and pleasant, although for some reasons, I cannot read the labels of several of them, this should be due to pdf-encoding. Compared to what has been done in the literature, it is shown here that conditioning lagrangian trajectories on the square of acceleration allows to use a gaussian description, amenable to tractable treatments of the statistical quantities. The article is especially well calibrated to be accessible to the entire physics community.

For these reasons, including a clear and pretty form accessible to a large community, and the mentioned newly observed gaussian behavior, I would suggest for a publication in Nature Communications, if the authors take into account my following remarks.

1. Why using a filtering of a^2 over its correlation time-scale? Would it not work directly with a^2 (i.e. using F a delta-function)?
2. I cannot read the labels of several figures, might be due to pdf encoding.
3. Is represented in fig. 2B the conditional pdfs of acceleration on the square of acceleration, and they are found close to gaussian. Is not it expected? Can it be expressed analytically, for a given pdf of acceleration?
4. Following this remark, eq. 2 asks for a Gaussian conditional pdf for velocity (and not acceleration), and such conditional pdfs are not displayed in the current manuscript. Are they really gaussian in the DNS? And so why representing conditional pdfs of acceleration?
5. I understand that authors take $f(\alpha)$ and C_α directly from DNS 'without further modeling', but at the very end, how the overall functionals depend on Reynolds number (and/or intermittent corrections)? Please comment on these important dependencies.

Reply to Reviewer #1

May 8, 2019

Dear Prof. Aluie,

Thank you very much for reviewing our manuscript. In the following, we provide point-by-point replies to the points raised in your report.

Reviewer: The paper investigates the problem of intermittency observed in turbulent flows along Lagrangian particle trajectories. Understanding the Lagrangian nature of turbulence is a fundamental problem in the field, with a wide range of applications such as modeling chemical combustion, describing nutrient transport in the ocean, predicting pollutant dispersal in the atmosphere, etc ...

The paper can be viewed as providing a Lagrangian analogue of Kolmogorov's Refined Similarity Hypothesis framework, which is primarily an Eulerian description. Much of our current understanding of turbulent flows rests on Eulerian frameworks. There is widespread belief in the turbulence community that the underlying physics is ultimately Lagrangian in nature, yet a self-consistent and methodical translation of the main Eulerian results in turbulence to a Lagrangian formulation has remained elusive. The current work is an important contribution in this regard, which I believe is a promising framework to pursuing such a research program.

Reply: Thank you for this very positive assessment of our results and their possible implications.

Reviewer: While I believe the paper is suitable for publication in Nature Communications, I have a few comments and questions, some of which I believe are important.

I feel that the presentation in its current form is perhaps too specialized and does not cater to the wide background of the journal's readers. I would suggest the authors to revise their manuscript by focusing more on the physical implications of their work, dwell less on the mathematical technicalities/details, and spend more effort at explaining the relevance of various plots. I would suggest they consider a reader who works in turbulence but is perhaps not versed in the techniques used by the authors: why should s/he care about PDFs of increments, their heavy tails, log-normal distributions, velocity correlations, characteristic functionals, and discuss more the physical insight afforded by the

framework/model put forth by this work. For example, the first paragraph on page 4 illustrates how the paper can be more technical than is necessary. I believe these revisions will eventually benefit the paper and make it accessible to a wider audience. Some of my comments/questions below provide some more concrete suggestions.

Reply: Thank you for pointing out that the manuscript might be difficult to understand for a reader with a different background. Throughout the revised manuscript, we now provide additional information and interpretation of the presented quantities. We have also added subsections to the manuscript to distinguish the observational section of the manuscript from the sections in which the theoretical framework is developed and tested. Wherever possible, we have relegated mathematical details to the Methods section and the Supplementary Information. We hope you agree that these changes render the manuscript more accessible for a diverse readership.

Reviewer: Another important aspect of the paper which I would like the authors to address pertains to the coarse-graining time scale. What is special about 3-4 Kolmogorov times? In Kolmogorov's refined similarity framework, the coarse-graining length scale is that of the scale being analyzed. Here, it does not seem related to any of the quantities being analyzed (e.g. in Fig. 4c). Why is that?

Reply: Thank you for raising this important point. Indeed, this is one main difference compared to Kolmogorov's refined similarity hypothesis: in Kolmogorov's refined similarity hypothesis, the statistics of the flow on a certain scale are assumed to be universal depending on the dissipation rate averaged *over the given scale*. Contrary to this, we define the coarse-grained squared acceleration α independent of the quantity we want to describe, using a fixed coarse-graining time scale Θ .

We think that both approaches can be justified. Ultimately, they are validated by the observation of universal conditional statistics in the data. In the Eulerian frame, conditional statistics on the scale-dependent average of the dissipation rate/energy transfer rate have indeed been observed to take a universal (Gaussian) form [1–4]. In our Lagrangian case, however, we observe universal Gaussian statistics on all scales when fixing the coarse-graining time scale Θ to a suitable value. Our interpretation is as follows: The goal of discriminating Lagrangian trajectories with respect to their coarse-grained acceleration is to separate them into sub-ensembles based on the degree of small-scale turbulence they encounter. We find that the acceleration is in principle a suitable quantity to achieve this, but coarse-graining is needed to clarify how representative the acceleration event is. If the coarse-graining time scale is too small, we rely on a criterion based on a single instant in time, which may not represent the degree of the flow region well. On the other hand, if the coarse-graining time scale is too large, the discrimination into sub-ensembles becomes less sharp, because over

longer times Lagrangian tracers probe regions of mild *and* strong turbulence. Therefore, we search for an intermediate time scale, which is “just right” to discriminate flow regions into mild and strong turbulence. Empirically, we find that this time scale is comparable to the time scale over which the acceleration components are significantly correlated. Overall, this finding can be considered a simplification compared to the refined similarity hypothesis, which allows for an intuitive interpretation of our framework: The Lagrangian particles are separated into groups corresponding to different regions of the flow, from quiescent areas up to highly turbulent activity.

With regards to the conceptual implications, the fixed time scale also enables the generalization of our framework to the characteristic functional: Discriminating the particles with respect to α yields one single decomposition of the ensemble of particles. Since in each sub-ensemble, the observation of Gaussianity is scale-independent, we are able to generalize the modeling from single statistical quantities such as velocity increment PDFs to the entire stochastic process, which we describe by a superposition of Gaussian characteristic functionals. Thus we are able to construct a comprehensive model of single-particle statistics – an important conceptual generalization of previous approaches.

Of course, fixing the time scale introduces the free parameter Θ into the framework, which has to be chosen in a reasonable way. Based on your comment and the comments of the other reviewers, we have clarified this point in the revised manuscript. Furthermore, we have added material to document the dependence of our results on the coarse-graining time scale to the Supplementary Information.

Reviewer: How persistent is the “persistent acceleration” put forth by the authors? Fig. 1 suggests it persists over ~ 10 Kolmogorov times. Is this the case?

Reply: Thank you for raising this question. Also in the light of the comments of the other reviewers, we realized that some aspects of the concept of “persistent accelerations” may be prone to misunderstanding. The notion can be interpreted in two different ways: in your question, you seem to refer to single events during which the coarse-grained squared acceleration α retains a certain value for a long period of time. Such an event can be seen in Fig. 1 of the manuscript. We show this figure to illustrate how a coarse-graining of the acceleration can even out the fluctuations a particle experiences while being trapped within the same small-scale structure. Furthermore, it is a typical feature of our framework that the highest values of α appear to select very specific “particle trapping” events, an example of which can be seen in that figure. In this example, the event indeed lasts for about ten Kolmogorov time scales. However, the duration of the event is not what our notion of “persistence” refers to.

Contrary to these single “persistent events”, the interpretation that we would like to put forward refers to the coarse-graining procedure. As elaborated above, we introduce a temporal coarse-graining of the squared acceleration in order to

capture the local degree of small-scale turbulence as opposed to capturing a single snapshot of acceleration. In other words, we measure the order of magnitude of the acceleration that *persists* over several Kolmogorov time scales. In that sense, the persistence is given by our coarse-graining time scale Θ . In the revised manuscript, we now introduce the notion in this way and distinguish more clearly the aspects which are related to the coarse-graining from the aspects which refer to long-lasting events.

Reviewer: The quantity α is introduced without much motivation. The reader has to wait until the last paragraph of the paper to see that it is analogous to the measure of dissipation. If this is what inspired the authors to focus on α , it would benefit the reader to discuss it soon after/before α is introduced in eq. (1).

Reply: Thank you for pointing out that this motivational point would be helpful to the reader at an earlier point of the manuscript. We have moved the discussion of the analogy to the refined similarity hypothesis to the point where the quantity α is introduced. Furthermore, as mentioned above, we have added a discussion on the choice of the coarse-graining time scale to the manuscript.

Reviewer: The angular brackets first appear in Fig. 2 without being defined. The appendix defines them as an ensemble average. Please mention that in the main paper. In the appendix, mention how the ensemble average is taken in practice. Is it a time average and, if so, how do you justify/ensure statistical independence?

Reply: Thank you for pointing out that this definition is missing. Both the definition of the ensemble average and of the conditional ensemble average have been added to the position where they are first used. The ensemble average is taken over a set of 2 to 16 million tracer particles. Additionally, a time average over the simulation is performed, covering about 7 integral time scales (or 24 for $R_\lambda \approx 210$). For example, statistics of the velocity increment are computed for increments centered around snapshots that are evenly spaced throughout the total simulation time. Of course, the samples adjacent in time are not statistically independent. However, by taking into account a large ensemble over a long time, we ensure a sufficiently large number of statistically independent samples. This information has been added to the SI. To further improve the statistical quality of our results, we have performed additional simulations, one of which we now use as the main data set in the manuscript.

Reviewer: In Fig. 2a, it would help if the velocity time series corresponds to the same α values used in Fig. 2b and Fig. 3. Currently, Fig. 2a only shows velocity at 3 different values of α but each value shows 4 different u-series. Why is that and does the color legend correspond to that in Fig. 2b?

Reply: Thank you for raising this question. The color code of Fig. 2a in the original manuscript did indeed correspond to that of Fig. 2b and 3. In order to illustrate the statistical behavior of the conditional velocity time series, we decided to show multiple realization for each value of α . Since showing all of these trajectories would have cluttered the plot, we reduced to a smaller subset of values of α .

In the revised version of the manuscript, we have rearranged the plots. The new Fig. 2 shows the PDF of α along with an illustration of typical velocity time series for three different values of α . These are displayed in separate panels for clarity. Their corresponding value of α is indicated in the plot of the PDF. Furthermore, the color codes of the Fig. 2, 3 and 4 are now matching.

Reviewer: On page 3, bottom of the left column, the authors say "At high values of α , conditional velocity correlation functions exhibit a second local maximum after a few Kolmogorov time scales, which indicates oscillatory motion." The statement is somewhat vague, at least to me. I would expect oscillatory motion for any α , perhaps at varying amplitudes. This certainly appears to be the case in Fig. 1a. So how am I to interpret the secondary maximum in the conditional velocity correlation function for $\alpha = 61$ but not at lower values of α ?

Reply: We thank the reviewer for pointing out that this point needs further explanation. It is true that purely oscillatory motion at different amplitudes or, alternatively, at different frequencies would lead to different α , each corresponding to a certain amplitude or frequency, respectively. However, the range of possible geometries of Lagrangian trajectories in a fully developed turbulent flow is much more diverse, among which oscillations are only one possible realization.

What we suggest is that the largest values of α correspond to a clearly defined type of event, a particle being trapped in an intense small-scale structure. This hypothesis comes from the observation of oscillations in typical velocity time series for high values of α (Fig. 2d in the revised manuscript). This is furthermore supported by the fact that α is similar to the quantity used by Biferale et al. [5] in order to select particle trapping events. Intuitively, it makes sense that the highest values of α can only be reached when particles retain a persistently high acceleration magnitude, a typical feature of spiraling motion. The fact that the correlation function exhibits a second local maximum supports this idea and furthermore indicates that the particle trapping events have a clearly defined frequency.

For lower α , we expect a much broader range of distinct Lagrangian trajectories that lead to the same value of α . For example, a high amplitude, low frequency oscillation of the velocity could lead to the same value of α as a low amplitude, high frequency oscillation. The same is true for particles that transition between different regions of the flow, for example a particle exiting a vortex trapping event. This is why one should not expect clearly defined oscillations

for lower values of α .

Reviewer: On page 4, the statement: "... Lagrangian intermittency can be perceived as a consequence of the statistical mixture of different regions of the flow, each characterized by a particular persistent acceleration." I believe this is insightful, but I have a few questions. First, is the use of the word "persistent acceleration" justified (related to question 1 above)? Over what time periods is this persistence observed and how does it compare to the coarse-graining time scale?

Reply: Thank you again for raising this important point. As explained above we came to realize that some aspects of our notion of "persistent acceleration" may be prone to misunderstandings. We stress again that the notion does not refer to certain "persistent events" but to the coarse-graining procedure. Due to the coarse-graining, each sub-ensemble is characterized by the acceleration that *persists* in time, i.e. that contributes most dominantly to the coarse-grained squared acceleration α . So the duration of persistence is in fact given by the coarse-graining time scale. In the revised manuscript, we are now explaining this interpretation more clearly.

Reviewer: Second, why is there sensitivity to the coarse-graining time scale and how should I interpret this dependence (related to the main comment above)? It is not clear if the coarse-graining time scale should be made longer or shorter as the Reynolds number is increased. Is it possible to present a plot of the optimal filter time scale as a function of Re ?

Reply: Regarding your question on the interpretation of sensitivity of our results on the coarse-graining time scale, please refer to our reply above. Although there are ways to argue for certain values of Θ , ultimately it is a free parameter of the theory. In the following, we address the idea of an optimal Θ , as well as a possible dependence of this optimal value on the Reynolds number. We have also added additional material to the Supplementary Information.

To address your point, we have analyzed a larger set of DNS, including both previously generated [6] and new data sets. One possible method to determine an optimal Θ is to consider the predicted acceleration flatness and to choose Θ such that it matches the DNS value. In the SI, we now present this definition of an optimal Θ . While we do observe a trend that the optimal Θ value decreases with increasing Reynolds number for some data sets, we also found clear exceptions. We therefore additionally tried to infer any possible influence that spatial resolution (i.e. $k_M \eta$) or temporal resolution (i.e. the CFL number) have on the results, and they do not seem to be strongly influenced by either. Furthermore we note that this particular criterion of the optimal Θ is very delicate, and Fig. S6 in the SI suggests that the notion of "optimal" is not very rigid. Considering our results, it seems likely that a correct parametrization of the optimal value of Θ is a considerable effort in itself, while at the same time we do find very

reasonable predictions for a range of Θ choices. In order to maintain clarity of scope, we do not present this detailed analysis in the paper.

Reviewer: There is a typo in a sentence in the last paragraph of page 5. Too many "but".

Reply: Unfortunately, we fail to see the problem with this sentence. Can you please elaborate?

Based on these replies and the extensive revision of the manuscript, we hope that you will find the manuscript suitable for publication. Once more, thank you very much for your constructive review.

Yours sincerely,
Michael Wilczek (on behalf of all authors)

References

- [1] Homann, H., Schulz, D. & Grauer, R. Conditional Eulerian and Lagrangian velocity increment statistics of fully developed turbulent flow. *Phys. Fluids* **23**, 055102 (2011).
- [2] Lawson, J. M., Bodenschatz, E., Knutsen, A. N., Dawson, J. R. & Worth, N. A. Direct assessment of Kolmogorov's first refined similarity hypothesis. *Phys. Rev. Fluids* **4**, 022601 (2019).
- [3] Gagne, Y., Marchand, M. & Castaing, B. Conditional velocity pdf in 3-D turbulence. *J. Phys. II France* **4**, 1-8 (1994).
- [4] Naert, A., Castaing, B., Chabaud, B., Hébral, B. & Peinke, J. Conditional statistics of velocity fluctuations in turbulence. *Physica D* **113**, 73 - 78 (1998).
- [5] Biferale, L., Boffetta, G., Celani, A., Lanotte, A. & Toschi, F. Particle trapping in three-dimensional fully developed turbulence. *Phys. Fluids* **17**, 021701 (2005).
- [6] Lalescu, C. C. & Wilczek, M. How tracer particles sample the complexity of turbulence. *New J. Phys.* **20**, 013001 (2018).

Reply to Reviewer #3

May 8, 2019

Dear Reviewer,

Thank you very much for reviewing our manuscript. In the following, we provide point-by-point replies to the points raised in your report.

Reviewer: This work proposes a framework to reduce the strongly non-Gaussian statistics of Lagrangian turbulence using sub-ensembles to Gaussian statistics and to predict from this statistical quantities. More specifically, the authors show that the acceleration probability functions become Gaussian once they are conditioned on an averaged acceleration. Next, using Gaussian characteristic functionals of this conditioned quantity they reconstruct the full characteristic functional that allow them to correctly predict for example the complete velocity increment statistics.

It is true that understanding the intermittent nature of turbulent flows is still an open problem. It is also true that Kolmogorov's and Obukhov's refined similarity hypothesis was an important milestone from the Eulerian point of view. Their coarse-grained energy dissipation rate disentangles sets with different scaling properties. The present manuscript proposes a mechanism to achieve the same from the Lagrangian point of view and to build on this a formalism to predict turbulence properties.

Reply: Thank you very much for your review and this accurate assessment of our paper and its importance.

Reviewer: The present paper is well written and well structured. My concern about the present paper is that the proposed framework appears as a clever modelling although the authors explicitly reject this kind of criticism.

Reply: We thank the reviewer for raising this important point. Let us clarify from the outset that we do not explicitly reject the statement that our theoretical framework contains modeling aspects. Our framework is indeed a model for Lagrangian single-particle statistics. What we mean to stress in our manuscript is

1. that we have developed a self-consistent theoretical framework (which you also may call a model) of Lagrangian single-particle statistics. This framework crucially relies on the fact that we have identified a quantity (the coarse-grained squared acceleration) which separates the set of Lagrangian trajectories into quasi-Gaussian sub-ensembles, and
2. that this framework captures a number of important aspects of Lagrangian turbulence, including Lagrangian intermittency.

Reviewer: The main quantity called 'persistent acceleration' α is the acceleration averaged along a tracer trajectory over a short time lag. This time lag is of the order of τ_η which is the correlation time of the tracer acceleration. Indeed, the authors find Gaussian statistics for the tracer acceleration once they condition it on its short time average. However, the condition α almost determines the acceleration (one point quantity) as the first is the (correlated) time average of the latter. Therefore all sets of acceleration (for different α) will have a very small standard deviations. The subsequent step of the proposed framework, the prediction of turbulence properties such as velocity increment statistics uses measured quantities (from they direct numerical simulations) such as the occurrence of α and the velocity auto-correlation. So that the these predictions resemble reconstructions.

The authors should discuss in more detail the crucial conditioning on a short time lag. What happens for instance if the time lag tends to zero?

Reply: To answer your question, let us start with elaborating in detail on your main concern, which we believe originates from a slight misconception of our approach. If we interpret your remark correctly, this concern arises from the fact that in Fig. 2b of the original manuscript, the conditioning quantity $\alpha(t)$, a time-average of the squared (vectorial) Lagrangian acceleration, is very similar to the conditioned quantity $a(t)$, a component of the Lagrangian acceleration. It is true that if one chose $\alpha(t) = a(t)$ instead, $f(\alpha)$ would simply be the acceleration PDF, and the conditional acceleration PDFs would reduce to delta functions:

$$f(a|\alpha) = \delta(a - \alpha). \quad (1)$$

The acceleration PDF $f(a)$ could then be written as a simple superposition of those delta functions. If one additionally loosened the assumption of zero mean in the sub-ensembles, one could reach a precise reconstruction of the acceleration PDF. We agree this process would yield no physical insight, and we stress that this is not how we obtain the acceleration PDF.

Let us elaborate on this in more detail. First of all, our approach relies on separating the set of Lagrangian trajectories into quasi-Gaussian sub-ensembles, which is motivated in more detail below. In contrast to the situation discussed above, we assume zero mean in the individual sub-ensembles, a reasonable assumption given that our conditioning quantity α only depends on the acceleration magnitude, which is also backed up by our observations. Contrary to the

Figure 1: **Non-standardized conditional acceleration PDFs.** The conditional PDFs on α all display close-to-Gaussian form, but their variance grows with α (compare Fig. 3b of the revised manuscript). The unconditional PDF is shown in red for comparison.

Reviewer’s intuition, the individual sub-ensembles have widely varying standard deviations. To demonstrate this, we have added a figure to the revised manuscript (Fig. 3b), which shows the conditional acceleration variance as a function of α . As dimensionally expected, we find a power-law behavior with an exponent close to 1. Hence, our non-Gaussian acceleration PDF is a superposition of zero-mean Gaussian acceleration PDFs of varying width. To further illustrate this point, Fig. 1 shows a version of Fig. 3a in the manuscript, in which the conditional PDFs have not been standardized.

To expand on your remark on a more formal level, let us consider an intermediate case between the extreme example above (i.e. $\alpha(t) = a(t)$) and the approach taken in our framework: Let us set $\alpha(t) = |\mathbf{a}(t)|$, i.e. equal to the magnitude of the acceleration vector. This yields the same conditional PDFs as taking the limit $\Theta \rightarrow 0$ in our framework, i.e. $\alpha(t) = |\mathbf{a}(t)|^2$, because $|\mathbf{a}(t)|$ and $|\mathbf{a}(t)|^2$ contain the same information. We want to calculate the conditional PDF $f(a | |\mathbf{a}|)$ of a component a on the magnitude $|\mathbf{a}|$ using spherical coordinates. In the following, each f denotes the PDF of its argument. The statistical isotropy of the vector \mathbf{a} implies that its PDF can be written as a function of the magnitude only:

$$f(\mathbf{a}) = G(|\mathbf{a}|). \quad (2)$$

Thus, we obtain:

$$f(a \mid |\mathbf{a}|) = \frac{f(a, |\mathbf{a}|)}{f(|\mathbf{a}|)} \quad (3)$$

$$= \frac{\int d^3 \mathbf{x} f(\mathbf{a} = \mathbf{x}) \delta(x_3 - a) \delta(|\mathbf{x}| - |\mathbf{a}|)}{\int d^3 \mathbf{x} f(\mathbf{a} = \mathbf{x}) \delta(|\mathbf{x}| - |\mathbf{a}|)} \quad (4)$$

$$= \frac{\int d\phi d\vartheta dr G(r) \delta(r \cos \vartheta - a) \delta(r - |\mathbf{a}|) r^2 \sin \vartheta}{\int d\phi d\vartheta dr G(r) \delta(r - |\mathbf{a}|) r^2 \sin \vartheta} \quad (5)$$

$$= \begin{cases} \frac{1}{2|\mathbf{a}|}, & \text{if } -|\mathbf{a}| < a < |\mathbf{a}| \\ 0, & \text{else.} \end{cases} \quad (6)$$

From (5) to (6) we have substituted the delta function (using $\vartheta \in (0, \pi)$):

$$\delta(r \cos \vartheta - a) = \begin{cases} \frac{\delta(\vartheta - \cos^{-1}(a/r))}{|r \sin(\cos^{-1}(a/r))|}, & \text{if } -r < a < r \\ 0, & \text{else.} \end{cases} \quad (7)$$

So the conditional PDF of a on $|\mathbf{a}|$ is a uniform PDF between $-|\mathbf{a}|$ and $|\mathbf{a}|$. This PDF is neither Gaussian, nor does it necessarily have “small” standard deviation, since the standard deviation grows with α .

In order to see that our results are in fact both non-trivial and insightful, let us also stress that we are not only considering the single-point acceleration statistics, but a rather comprehensive set of multi-scale statistics including the intermittent Lagrangian velocity increment statistics. Given its dependence on the time-lag, capturing this quantity goes significantly beyond modeling, say the single-time acceleration PDF only. This demonstrates how our approach goes beyond just pure reconstruction.

Let us now elaborate about our rationale for choosing the coarse-grained squared acceleration to discriminate the set of Lagrangian trajectories into quasi-Gaussian sub-ensembles. The goal of discriminating Lagrangian trajectories with respect to their coarse-grained acceleration is to separate them into sub-ensembles based on the degree of small-scale turbulence they encounter. We find that the acceleration is in principle a suitable quantity to achieve this, but coarse-graining is needed to clarify how representative the acceleration event is. If the coarse-graining time scale is too small, we rely on a criterion based on a single instant in time, which may not represent the degree of the flow region well. On the other hand, if the coarse-graining time scale is too large, the discrimination into sub-ensembles becomes less sharp, because over longer times Lagrangian tracers probe regions of mild *and* strong turbulence. Therefore, we search for an intermediate time scale, which is “just right” to discriminate flow regions into mild and strong turbulence. Empirically, we find that this time scale is comparable to the time scale over which the acceleration components are significantly correlated.

For completeness, we have added a section to the Supplementary Information, where we discuss the choice of Θ . As one can see, the case $\Theta \rightarrow 0$ neither

yields Gaussian conditional statistics, nor does it yield a good reconstruction of the acceleration PDF. Instead, only when coarse-graining at the order of a few τ_η , the conditional acceleration PDF, increment PDFs at various time lags and velocity PDF become Gaussian. Since neither of these quantities can be directly “reconstructed” from α , we think that this reveals an intrinsic feature of Lagrangian turbulence.

Finally, let us explain how also our mathematical framework goes beyond reconstruction. While it is true that accurately capturing the acceleration and increment statistics can be regarded as a reconstruction, we would like to emphasize that this is not achieved by design, but crucially depends on the non-trivial observation that conditioning on α decomposes Lagrangian statistics into Gaussian sub-ensembles. Since we have found a decomposition into zero-mean Gaussian distributions, only their second-order moments are needed in order to define them uniquely. These are captured by the conditional velocity auto-correlation functions. Using only the distribution of sub-ensembles and their correlations, we obtain a simple model that completely captures Lagrangian intermittency. We are able to describe the full range from small-scale extreme events to large-scale Gaussianity using only a single decomposition of the Lagrangian particles into groups of different turbulent activity. In this procedure, non-Gaussianity is achieved by combining ensembles with varying correlations.

In order to construct a unified framework for acceleration and increments on all scales and to allow for the prediction of additional single-particle quantities, we extend the assumption of Gaussianity from finite-dimensional quantities to the full stochastic process. In that sense, our framework is much more than a reconstruction. It models a wide range of quantities in a self-consistent way, i.e. kinematic relationships are obeyed. An example is given by the increment PDF equation discussed in the manuscript.

To conclude, let us summarize why we think that our results give deep physical insight into the Lagrangian statistics of turbulent flows:

1. As elaborated on above, the observation of Gaussianity in the conditional acceleration PDF is not expected from simple theoretical considerations. Instead, it is an intrinsic feature of Lagrangian turbulence, for which to date we can only give an explanation by physical intuition. This applies even more to the conditional increment PDFs, which are now presented in the Supplementary Information. We therefore think that this observation is a strong reduction of the complexity of Lagrangian statistics, featuring an accessible and illustrative physical interpretation.
2. Due to the fact that we do not assume a functional form for the model parameters, i.e. the PDF of α and the sub-ensemble velocity autocorrelation functions, but take these quantities directly from DNS data, one might call our framework in the current form a reconstruction. However, this reconstruction relies on the strong assumption of Gaussianity in each sub-ensemble. Thus the parameters of our model have far less degrees of freedom than the quantities it is able to predict. Once theoretical models

of the PDF of α and the sub-ensemble velocity autocorrelation functions become available, our framework becomes fully predictive.

Reviewer: From a presentational point of view it would be instructive to present the flatness of the pdfs of the conditioned acceleration in FIG. 2 as a measure of the deviation from Gaussianity as a function of α .

Reply: Thank you for this suggestion to give a more quantitative picture of the presented data. To address this point we have included a plot of the conditional variance as well as the conditional flatness to the manuscript.

Based on these replies and the extensive revision of the manuscript, we hope that you will find the manuscript suitable for publication. Once more, thank you very much for your constructive review.

Yours sincerely,
Michael Wilczek (on behalf of all authors)

Reply to Reviewer #4

May 8, 2019

Dear Reviewer,

Thank you very much for reviewing our manuscript. In the following, we provide point-by-point replies to the points raised in your report.

Reviewer: This is a pretty and nicely written article, that attacks a difficult emblematic problem of non-equilibrium non-linear physics. It is a real pleasure to read the article, and references give a pretty much complete overview of the field. Figures are clear and pleasant, although for some reasons, I cannot read the labels of several of them, this should be due to pdf-encoding. Compared to what has been done in the literature, it is shown here that conditioning lagrangian trajectories on the square of acceleration allows to use a gaussian description, amenable to tractable treatments of the statistical quantities. The article is especially well calibrated to be accessible to the entire physics community.

For these reasons, including a clear and pretty form accessible to a large community, and the mentioned newly observed gaussian behavior, I would suggest for a publication in Nature Communications, if the authors take into account my following remarks.

Reply: We thank the reviewer for this very positive assessment of our results. Following a request of Reviewer 1, we have added additional explanations and interpretations of our results to the manuscript to cater an even wider audience.

Reviewer: Why using a filtering of a^2 over its correlation time-scale? Would it not work directly with a^2 (i.e. using F a delta-function)?

Reply: Thank you for raising the important question about the proper coarse-graining time scale. Our rationale of choosing the coarse-grained, squared acceleration is to discriminate the set of Lagrangian trajectories into quasi-Gaussian sub-ensembles based on the degree of small-scale turbulence they encounter. We find that the acceleration is in principle a suitable quantity to achieve this, but coarse-graining is needed to clarify how representative the acceleration event is. If the coarse-graining time scale is too small, we rely on a criterion based on a single instant in time, which may not represent the degree of the flow region

well. On the other hand, if the coarse-graining time scale is too large, the discrimination into sub-ensembles becomes less sharp, because over longer times Lagrangian tracers probe regions of mild *and* strong turbulence. Therefore, we search for an intermediate time scale, which is “just right” to discriminate flow regions into mild and strong turbulence. Empirically, we find that this time scale is comparable to the time scale over which the acceleration components are significantly correlated.

Regarding your suggestion of considering directly \mathbf{a}^2 , let us elaborate the following: Conditioning on \mathbf{a}^2 corresponds to conditioning on the magnitude $|\mathbf{a}|$, because it yields the same conditional PDFs, since $|\mathbf{a}|$ and $|\mathbf{a}|^2$ contain the same information. We want to calculate the conditional PDF $f(a | |\mathbf{a}|)$ of a component a on the magnitude $|\mathbf{a}|$ using spherical coordinates. In the following, each f denotes the PDF of its argument. The statistical isotropy of the vector \mathbf{a} implies that its PDF can be written as a function of the magnitude only:

$$f(\mathbf{a}) = G(|\mathbf{a}|). \quad (1)$$

Thus, we obtain:

$$f(a | |\mathbf{a}|) = \frac{f(a, |\mathbf{a}|)}{f(|\mathbf{a}|)} \quad (2)$$

$$= \frac{\int d^3\mathbf{x} f(\mathbf{a} = \mathbf{x})\delta(x_1 - a)\delta(|\mathbf{x}| - |\mathbf{a}|)}{\int d^3\mathbf{x} f(\mathbf{a} = \mathbf{x})\delta(|\mathbf{x}| - |\mathbf{a}|)} \quad (3)$$

$$= \frac{\int d\phi d\vartheta dr G(r)\delta(r \cos \vartheta - a)\delta(r - |\mathbf{a}|)r^2 \sin \vartheta}{\int d\phi d\vartheta dr G(r)\delta(r - |\mathbf{a}|)r^2 \sin \vartheta} \quad (4)$$

$$= \begin{cases} \frac{1}{2|\mathbf{a}|}, & \text{if } -|\mathbf{a}| < a < |\mathbf{a}| \\ 0, & \text{else.} \end{cases} \quad (5)$$

From (4) to (5) we have substituted the delta function (using $\vartheta \in (0, \pi)$):

$$\delta(r \cos \vartheta - a) = \begin{cases} \frac{\delta(\vartheta - \cos^{-1}(a/r))}{|r \sin(\cos^{-1}(a/r))|}, & \text{if } -r < a < r \\ 0, & \text{else.} \end{cases} \quad (6)$$

So the conditional PDF of a on $|\mathbf{a}|$ is a uniform PDF between $-|\mathbf{a}|$ and $|\mathbf{a}|$, in particular it is not Gaussian. This shows that if Θ is chosen too small, we do not obtain Gaussianity for the conditional acceleration PDF. By continuity, the same is true for conditional increment PDFs for small time lags.

In light of your comment and the comments of the other reviewers, we have now added a detailed discussion on the dependence of our results on the choice of the coarse-graining time scale to the Supplementary Information.

Reviewer: I cannot read the labels of several figures, might be due to pdf encoding.

Reply: Thank you for pointing out this technical problem, which is due to some system-dependent incompatibilities of font embedding in PDF figures. We have revised the figures to solve this problem.

Reviewer: Is represented in fig. 2B the conditional pdfs of acceleration on the square of acceleration, and they are found close to gaussian. Is not it expected? Can it be expressed analytically, for a given pdf of acceleration?

Reply: Thank you for raising this important question. The observation of Gaussianity is a crucial point of our manuscript and to our knowledge, there is no way to show this analytically. Let us look at different choices of Θ to answer your question:

1. As elaborated on above, in the case where we let $\Theta \rightarrow 0$, the conditional acceleration PDF can be calculated analytically, and it is a uniform distribution in the interval $[-\sqrt{\alpha}, \sqrt{\alpha}]$, so not at all Gaussian. Similarly, conditional increment PDFs for small time lags depart significantly from Gaussianity. Details on this limiting case have been added to the Supplementary Information.
2. In the opposite case, letting $\Theta \rightarrow \infty$, α will converge to the acceleration variance $\langle \mathbf{a}^2 \rangle$, independent of the particle and of the time of evaluation. Hence we expect the conditional PDFs to approach more and more the heavy-tailed unconditional acceleration PDF.
3. In between, in the case of finite Θ , we expect some intermediate shape of the PDF. Since in this case, the conditioning variable $\alpha(t)$ is a weighted average of squared accelerations in the vicinity of t , the conditional statistics will depend strongly on time correlations of the Lagrangian acceleration, which makes it much harder to treat analytically. The fact that we observe close-to-Gaussian statistics for all α at a well-defined Θ is therefore remarkable. Even more remarkably, we observe the same close-to-Gaussian form for the conditional increment PDFs at all time lags. We added a corresponding plot to the Supplementary Information as well.

Reviewer: Following this remark, eq. 2 asks for a Gaussian conditional pdf for velocity (and not acceleration), and such conditional pdfs are not displayed in the current manuscript. Are they really gaussian in the DNS? And so why representing conditional pdfs of acceleration?

Reply: Thank you for raising this question that requires some explanation. In equation (2) of the manuscript, we introduce the Gaussian characteristic functional in order to model the sub-ensemble statistics of the velocity time series $u(t)$ as a Gaussian process. We used $u(t)$ as the primary variable since it appears as the natural choice in the description of Lagrangian statistics. However, since we are dealing with a comprehensive description of the time series, using the

Figure 1: **Conditional velocity increment PDFs** for $\Theta = 3\tau_\eta$ and four different values of the time lag τ , for $R_\lambda = 350$. While the unconditional PDF (red line) transitions from a heavy-tailed distribution for small time lags to an almost Gaussian distribution for large time lags, the conditional PDFs (colored lines) are close to Gaussian on all scales. A Gaussian distribution (black, dashed line) is plotted for comparison.

acceleration time series $a(t)$ or the trajectory $x(t)$ would yield almost equivalent descriptions, as we explain in the following:

As the infinite-dimensional generalization of the characteristic function, the characteristic functional contains the full statistical information about the time series in continuous time. It is defined as [1]

$$\phi[\vartheta] = \left\langle \exp \left(i \int_{-\infty}^{\infty} dt \vartheta(t) u(t) \right) \right\rangle, \quad (7)$$

where $\vartheta(t)$ denotes a test function. This information also includes the time derivative $a(t)$ of $u(t)$. In fact, the characteristic functional $\psi[\eta]$ of the acceleration time series can be directly derived from it:

$$\psi[\eta] = \phi \left[\vartheta = - \frac{d\eta}{dt} \right]. \quad (8)$$

This can be shown by integration by parts [2]. Similarly, the full statistics of particle displacement is also contained in $\phi[\vartheta]$. Later on in the manuscript, we use this fact to derive finite-dimensional statistics of the displacement, velocity and acceleration all from the same functional $\phi[\vartheta]$.

In the observational part of the manuscript, we motivate making this strong assumption of a Gaussian process. However, we are only able to show projected, finite-dimensional quantities. We decided to show conditional acceleration statistics since they are the most extreme example of non-Gaussian, heavy-tailed statistics in Lagrangian turbulence. To further address your question, we show the conditional velocity increment PDFs at four different time lags in Fig. 1. They are indeed close-to-Gaussian like the acceleration. Also in the light of the questions of the other reviewers, we have added a discussion of these results to the Supplementary Information.

Reviewer: I understand that authors take $f(\alpha)$ and C_α directly from DNS “without further modeling” but at the very end, how the overall functionals depend on Reynolds number (and/or intermittent corrections)? Please comment on these important dependencies.

Reply: Thank you for raising this important point. Indeed, both the PDF of α as well as the conditional velocity autocorrelation function depend on the Reynolds number. To illustrate the Reynolds number dependence, we show in Fig. 2a how $f(\alpha)$ changes with R_λ . With increasing Reynolds number, the distribution widens spanning more values of α compared to its mean.

As α is a coarse-grained squared acceleration, we expect a statistics similar to the acceleration statistics in turbulence. To further look into the point on intermittency corrections that you raised, we can compare our results on α to the ones expected based on Kolmogorov’s classical theory from 1941 (K41) [3,4]. In this framework, dimensional arguments imply that the small-scale statistics should be universal at sufficiently high Reynolds number when expressed in

Figure 2: **PDF of α and its mean** for different Reynolds numbers and $\Theta = 3\tau_\eta$. **a**: The PDF of α widens with increasing R_λ , hence the observed extreme events become more extreme. **b**: In Kolmogorov units, the mean α grows with R_λ , which is consistent with intermittency-related deviations from the Heisenberg-Yaglom prediction (see text).

Kolmogorov units. In this context, it is important to note that the coarse-graining procedure leaves the mean of the time series unchanged. Therefore, the mean α corresponds to the acceleration variance: $\langle \alpha \rangle = \langle \mathbf{a}^2 \rangle$. As a consequence, the Heisenberg-Yaglom prediction for the acceleration variance [5–8], which is based on K41 arguments, makes statements also about $\langle \alpha \rangle$. The deviations from this prediction are illustrated in Fig. 2b, which shows that $\langle \alpha \rangle$ increases with Reynolds number when non-dimensionalized with Kolmogorov units, consistent with experimental findings on the acceleration variance [9]. This illustrates that intermittency, as expected, plays a role in the statistics of α .

The conditional autocorrelation functions $C_\alpha(\tau)$ display the same qualitative form for various Reynolds numbers, as shown in Fig. 3. For the highest values of α , they exhibit a second local maximum after a few Kolmogorov time scales. Some of them even display additional maxima or “shoulders” at the doubled time lag, confirming our interpretation of oscillations. Since with increasing Reynolds number, values of α become more extreme, we are able to show sub-ensembles corresponding to much larger multiples of its mean. However, clear oscillatory behavior appears to be restricted to the largest α .

In addition to our reply here, we now also discuss the performance of our framework for varying Reynolds number in more detail in the SI.

Based on these replies and the extensive revision of the manuscript, we hope that you will find the manuscript suitable for publication. Once more, thank you very much for your constructive review.

Yours sincerely,

Michael Wilczek (on behalf of all authors)

Figure 3: **Conditional velocity autocorrelation functions** for $R_\lambda = 210$ (left) and $R_\lambda = 509$ (right), for $\Theta = 3\tau_\eta$. As in Fig. 4 of the revised manuscript, they decay monotonically for small α and indicate oscillatory motion for large α .

References

- [1] Lumley, J. L. *Stochastic Tools in Turbulence* (Dover Publications, 2007).
- [2] Wilczek, M. Non-Gaussianity and intermittency in an ensemble of Gaussian fields. *New J. Phys.* **18**, 125009 (2016).
- [3] Kolmogorov, A. N. The Local Structure of Turbulence in Incompressible Viscous Fluid for Very Large Reynolds Numbers. *Dokl. Akad. Nauk SSSR* **30**, 301–305 (1941).
- [4] Kolmogorov, A. N. Dissipation of Energy in Locally Isotropic Turbulence. *Dokl. Akad. Nauk SSSR* **32**, 16–18 (1941).
- [5] Heisenberg, W. Zur statistischen Theorie der Turbulenz. *Z. Phys.* **124**, 628–657 (1948).
- [6] Yaglom, A. M. On the acceleration field in a turbulent flow. *C. R. Akad. USSR* **67**, 795–798 (1949).
- [7] Batchelor, G. K. Pressure fluctuations in isotropic turbulence. *Proc. Camb. Phil. Soc.* **47**, 359374 (1951).
- [8] Obukhov, A. M. & Yaglom, A. M. The microstructure of turbulent flow. *Prikl. Mat. Mekh.* **15**, 3–26 (1951).
- [9] La Porta, A., Voth, G. A., Crawford, A. M., Alexander, J. & Bodenschatz, E. Fluid particle accelerations in fully developed turbulence. *Nature* **409**, 1017–1019 (2001).

REVIEWERS' COMMENTS:

Reviewer #1 (Remarks to the Author):

The authors have addressed all of my comments. I believe it is an important contribution that deserves to be published in Nature Comm. I have two minor comments but do not need to see the manuscript again.

1) on page 2, please add an appropriate reference near the statement "... Eulerian refined similarity hypothesis, which play a key role in separating Eulerian statistics into simpler sub-ensembles."

2) in the Fig. 3 caption, please clarify which component of the acceleration is being used.

Reviewer #3 (Remarks to the Author):

The authors addressed all questions and remarks raised in the previous report.

Reply to Reviewer #1

June 7, 2019

Dear Prof. Aluie,
Thank you very much for reviewing our revised manuscript.

Reviewer: The authors have addressed all of my comments. I believe it is an important contribution that deserves to be published in Nature Comm. I have two minor comments but do not need to see the manuscript again.

1) on page 2, please add an appropriate reference near the statement "... Eulerian refined similarity hypothesis, which play a key role in separating Eulerian statistics into simpler sub-ensembles."

Reply: Thanks for your suggestion. We have added references to the original works by Kolmogorov and Obukhov.

Reviewer: 2) in the Fig. 3 caption, please clarify which component of the acceleration is being used.

Reply: Thanks for raising this point. Because we are investigating isotropic turbulence, all three velocity components exhibit the same statistics. We have added a clarifying remark to the manuscript.

Once more, thank you very much for your constructive feedback on our manuscript.
Yours sincerely,

Michael Wilczek (on behalf of all authors)